

**Variation of key elements in soils and plant tissues in subalpine forests**
**of the northern Rocky Mountains, U.S.A.**
David P. Pompeani[1], Kendra K. McLauchlan[1], Barrie V. Chileen[1], Kyra D. Wolf[2], Philip
E. Higuera[2,3]
[1]Department of Geography, Kansas State University, Manhattan, KS 66506
[2]Systems Ecology Intercollegiate Graduate Program, University of Montana, Missoula,
MT 59812
[3]Department of Ecosystem and Conservation Sciences, University of Montana, Missoula,
MT 59812
**Abstract**
The essential elements for the structure and function of forest ecosystems are
found in relatively predictable proportions in living tissues and soils; however, both the
degree of spatial variability in elemental concentrations and their relationship with
wildfire history are unclear. Quantifying the association between nutrient concentrations
in living plant tissue and surface soils within fire-affected forests can help determine how
these elements contribute to biogeochemical resilience. Here, we present elemental
concentration data (C, N, P, K, Ca, Mg, S, Fe, Mn, Zn) from 72 foliar and 44 soil samples
from a network of 15 sites located in the fire-prone subalpine forests of the northern
Rocky Mountains, U.S.A. Plant functional type is strongly correlated with carbon (C) and
nitrogen (N)— C concentrations are highest in coniferous needles, and N concentrations
are highest in broadleaved plant species. The average N/P ratio of foliage among samples
is $9.8 \pm 0.6$ ($\mu \pm 95\%$ confidence). This suggests that N is the limiting nutrient for these
plants, however several factors can complicate the use of N/P ratios to evaluate nutrient
status. Average C concentrations in organic soil horizons that were burned in regionally
extensive fires in 1910 or 1918 CE are lower than those from sites that burned prior to
1901 CE ($p<0.05$). This difference suggests that wildfires reduced the pool of soil C and
that the legacy of these fires can be measured a century later. Our results help aid in
modeling how changing wildfire regimes will influence biogeochemical cycling in
subalpine forests.
**1. Introduction**
Although living plant tissue is primarily composed of carbon (C), there are
approximately 20 other elements that are necessary for biochemical reactions and growth.
These other key elements—including nitrogen (N), phosphorus (P), potassium (K),
calcium (Ca), magnesium (Mg), and sulfur (in the form of $SO_4$)— play important roles in



regulating terrestrial ecosystem processes. The quantities of these elements are controlled
by a number of factors including bedrock parent material, soil composition (Kramer et
al., 2017), vegetation type (Hu et al., 2001), and climate (Campbell et al., 2009). It is
generally thought that plant tissue and soil nutrient concentrations can provide
information about element limitation to growth (Wardle et al., 2004;Boerner,
1984;Schreeg et al., 2014). However, relatively few studies have analyzed the
concentrations of a large suite of elements in both plant and soil samples from one region.
48         Plant traits, such as biochemical, physiological, and anatomical features measured
at the individual level (Violle et al., 2007), reflect the outcome of evolutionary processes
responding to environmental constraints (Valladares et al., 2007). Traits determine how
primary producers respond to abiotic and biotic environmental factors and influence a
host of other ecosystem services (Kattge et al., 2011). Plant functional type (PFT) models
capture a substantial fraction of the observed trait variation across plant species. A
relatively small number of PFTs (5-20) have been used to represent the functional
diversity of >300,000 documented plant species on Earth in global vegetation models
(Kattge et al., 2011). Growth-limiting elements in foliar material derived from soils, such
as phosphorus, make nutrient concentrations among the most plastic of plant traits
(Wright et al., 2004) because the availability of these nutrients varies spatially as a result
of parent bedrock composition. Nutrient concentrations in plants are influenced by a
number of other factors, including the age of the tissue and overall age of the plant
(Turner et al., 1977;Reich et al., 1992;Schreeg et al., 2014;Luo et al., 2017). Additionally,
sunlight and location on a tree branch can also affect nutrient concentration in leaves
(Schreeg et al., 2014), and weather-related factors such as precipitation and temperatures
can affect nutrient levels on a seasonal basis (Turner et al., 1977). Therefore, samples of
plant tissues taken over a short time period for several species across one region are
particularly useful for differentiating factors that influence foliar nutrient concentrations
and subsequent bioavailability of nutrients in leaf litter and upper soil horizons (Qualls
and Haines, 1991;Taylor et al., 1989).
69         High severity fires can profoundly impact the cycling of C, N, and other nutrients
in plants and soils (Chen et al., 2017;Fletcher et al., 2014;Certini, 2005;Dunnette et al.,
2014), and could therefore affect the ability of the forest to regenerate to pre-fire
conditions (Smithwick, 2011;McLauchlan et al., 2014). Several mechanisms determine
how forests respond to fire, and how cycling of C, N, and other elements subsequently
change (Pellegrini et al., 2018;Certini, 2005). For example, combustion of vegetation and
soils from frequent, low-intensity burning can lead to the loss of C and N to the
atmosphere (Pellegrini et al., 2014;Reich et al., 2001;Deluca and Sala, 2006;Yelenik et
al., 2013). Fires also release nutrients from plant tissues as ash, which can potentially
increase post-fire vegetation growth through soil nutrient enrichment (Boerner et al.,
2009;Hurteau and Brooks, 2011). Over longer time scales of centuries to millennia,
ecosystem modeling informed with paleofire history records suggest that changing fire



regimes can have substantial and long-lasting impacts on C and nutrient cycling
(Hudiburg et al., 2017;Kelly et al., 2016).
Here we provide data from 72 leaf and 44 soil samples from a network of 15 sites
in the coniferous subalpine forests of the northern Rocky Mountains, U.S.A. (Fig. 1,
Table 1). The samples were collected during the middle of the growing season from sites
with similar elevation, climate, and bedrock geology, but with varying fire histories over
the 20$^{th}$ century. Our approach isolates local plant and soil forming processes that affect
the distribution of key elements, and compares sites that are reported to have burned in
extensive fires in 1910 and 1918 CE to those that have not burned since before 1901 CE
(Morgan et al., 2014). We evaluate nutrient status and characterize patterns of element
distribution within and among the leaves of several tree and shrub species in the context
of associated bioavailable nutrient data from upper soil horizons. We use statistical
methods to infer the transfer and distribution of key elements in soil and vegetation.
**2. Methods**
*2.1 Study area*
The study sites are located between 1623-2011 m above sea level in the northern
Bitterroot Mountains along the border of Idaho and Montana, U.S.A. (Fig. 1). The
vegetation at all sites is subalpine forest dominated by subalpine fir (*Abies lasiocarpa*),
Engelmann spruce (*Picea engelmannii*), lodgepole pine (*Pinus contorta*), and mountain
hemlock (*Tsuga mertensiana*). Of the 14 sites, seven experienced wildfire in 1910 or
1918 CE, while the other seven have no indication of fire activity back to 1901 CE
(Morgan et al., 2014)(Table 1). The regional climate is classified as modified maritime
with warm, dry summers and cool, wet winters. The study region is underlain with
partially metamorphosed argillites of the Belt Basin Supergroup, known as Belt meta-
sedimentary rocks. Glacial till and shallow entisols overlay the bedrock (Sasich and
Lamotte-Hagen, 1989).
*2.2 Fieldwork*
Soil cores, approximately 2 cm in diameter and 25 cm deep, and foliage samples
were collected at 15 sites along a 100-km northwest to southeast transect (46.78°-47.36°
N; 114.76°-115.57° W) from 12 July to 25 August, 2017 CE (Fig. 1). At each soil core
site, except for Hoodoo, associated leaf samples were collected from nearby trees and
shrubs, avoiding new growth and juvenile trees. The majority of the foliar samples
consist of coniferous needles from eight species (n=68): *Abies lasiocarpa*, *Larix*
*occidentalis*, *Pinus albicaulis*, *Pinus contorta*, *Picea engelmannii*, *Taxus brevifolia*,
*Thuja plicata*, and *Tsuga mertensiana*. We also collected two samples each of the



broadleaf species green alder (*Alnus viridis*; n=2) and false huckleberry (*Menziesia*
*ferruginea;* n=2). Foliar samples were taken directly from living vegetation located
approximately 2 m above the soil surface. Soil cores were sampled by horizons: 'litter'
was unconsolidated plant material from the surface; 'organic' soil was identified visually
and sampled at a depth between approximately 0-10 cm; the 'mineral' soil layer was
sampled between 10-25 cm below the surface depending on overall soil depth. Samples
were stored in airtight plastic containers for transport back to the laboratory.
*2.3 Laboratory analyses*
Samples were dried, homogenized, and sent to the Department of Agronomy Soil
Testing Lab at Kansas State University. Foliar samples were weighed into 50 ml Kimax
digestion tubes for elemental analysis. Boiling chips and 5 ml of nitric acid at 50%
strength were added to the tubes, which were covered with plastic wrap to react
overnight. The following day, 5 ml of perchloric acid was added to predigested plant
material and tubes were then placed on a cold techator digestion block. Temperatures
were set to 200°C and digests were heated for approximately three hours or until white
fumes appeared and the acid was clear (and colorless when cooled to room temperature)
(Gieseking et al., 1935). Tubes were diluted to 25 or 50 ml with deionized water and
mixed by inverting twice. Each digested sub-sample was then analyzed for P, Mg, K, Ca,
iron (Fe), manganese (Mn), zinc (Zn), and $SO_4$ using a Varian Model 720-ES ICP Optical
Emission Spectrometer. Analytical precision ($\sigma$) for element analysis was <0.0001%.
Base cations (Ca, Mg, K, and Na) were extracted from soil sub-samples using 1
M ammonium acetate (pH 7.0). The resulting supernatant was then analyzed using a
Varian Model 720-ES ICP Optical Emission Spectrometer. Trace elements (e.g. Mn, Zn,
etc.) were not measured on soil sub-samples. Soils were not analyzed for P.
Total carbon and nitrogen concentrations (weight %) in both plant and soil sub-
samples were measured using a LECO CN 2000 combustion analyzer. Analytical
precision ($\sigma$) was approximately 0.06% for carbon and 0.006% for nitrogen.
*2.4 Statistics*
To compare elemental concentrations among different sample sites, we used box
plots, unpaired t-tests (assuming unequal variance), principal components analysis
(PCA), and Pearson correlation coefficients. We calculated principal components using
the ade4 package in R (Dray and Dufour, 2007). Correlation tables were calculated using
the R base stats package (R Core Team).
**3. Results and discussion**



### 3.1 Patterns in soil and foliar elements

Soil C and N concentrations were highest on average (34.94% and 1.32%, respectively) within litter material and declined with depth through organic (18.30% and 0.84%) and mineral (6.05% and 0.29%) soil horizons (Table S1, Fig. 2). The concentration of other nutrients such as K, Ca, and Mg were highest on average within the litter (0.016%, 0.115%, and 0.012%, respectively) and organic soil horizons (0.019%, 0.104%, and 0.013%), and decreased in the mineral soil horizon (0.009%, 0.046%, 0.007%). Na concentrations displayed the opposite pattern and increased from litter (<0.001%) to mineral (0.001%) soil horizon.

The pattern of five elements in the soil— C, N, Ca, K, and Mg—correspond with the dominant role of decomposing plant material controlling the concentration of nutrients in forested soils (Aerts and Chapin, 1999). These results are consistent with a global compilation of 10,000 soil profiles that indicate ranking of nutrient concentrations from shallow to deep (Jobbagy and Jackson, 2001) (Fig. 2, Fig. S1). The decline in the bioavailable K, Ca, and Mg in the mineral horizon suggests removal by mineralization or microbial immobilization (Qualls et al., 1991;Qualls, 2000).

Foliage contained C concentrations between 47-52%, with the exception of *Larix occidentalis*, *Alnus viridis,* and *Menziesia ferruginea,* which had slightly lower concentrations of C between 43-47% (Fig. 3, Table S2). N concentrations were generally low (0.75-1.89%) in coniferous needles, and high in leaves of the broadleaved species *Alnus viridis* and *Menziesia ferruginea* (2.5-3.5%). P, K, and Ca concentrations were variable within and between tree and shrub species, with no clear pattern differentiating needle and broadleaf species (Fig. 3).

There was considerable variability in trace element concentrations among samples of foliar material (Fig. 3). Mg was below 0.20% in all leaf material, except for *Alnus viridis* (0.35%) and *Menziesia ferruginea* (0.50%). Fe, Mn, and Zn concentrations were highly variable within and between species. *Menziesia ferruginea* contained high concentrations of Mn (0.48%) and Zn (0.01%). $SO_4$ concentrations ranged between 0.01-0.15% for all foliar material, except for higher values found in *Alnus viridis* (0.18%).

Element concentrations did not vary with site latitude or elevation likely as a result of the constricted latitude and elevation range of the study sites. Soil and foliar N, K, and Mg were all positively correlated ($p<0.001$) with each other ($r=0.42-0.75$) (Figs. S1, S2, S3, and S4). In soils, Ca was positively correlated ($p<0.01$) with Mg (0.74), K (0.54), N (0.47), and C (0.39), whereas in plant tissues Ca was poorly correlated ($p>0.05$) with P (-0.21), K (-0.21), and Mg (0.15), C (0.05), and N (0.02). Foliar C was negatively correlated ($p<0.05$) with Mg (-0.53), N (-0.41), P (-0.38), and K (-0.30). In soils, Na was inversely related ($p<0.05$) to both C (-0.52) and N (-0.42). In vegetation, Zn was correlated ($p<0.05$) with Mn (0.54), K (0.33), and C (-0.26). Trace elements Fe and Mn were weakly correlated ($p>0.05$) with C, N, P, K, and Mg in plant tissues.



A multivariate analysis of elemental concentrations clearly separated foliar (n=72)
from litter/soil samples (n= 44) (principal component analysis [PCA]; Fig. 4). Axis 1 of
the PCA explained 58.7% of total variation in the soil and foliar elements. The
concentrations of several elements were negatively loaded on Axis 1: C (-0.905), K (-
0.809), Ca (-0.791), Mg (-0.799), and N (-0.706). C/N was correlated to Axis 1 (-0.535)
but more weakly than C, N, and the base nutrients. Axis 2 explained 22.4% of the
variation and was primarily driven by N and Mg (0.577, 0.381, respectively), and was
strongly influenced by the high foliar N concentrations of *Alnus viridis* (n=2). Foliage
samples from different plant species were dispersed along Axis 2.
Leaf carbon, nitrogen, and phosphorus content are considered important
biochemical traits in PFT modeling, as seen in the TRY plant database (Kattge et al.,
2011). We measured additional elemental concentrations to understand inter- and intra-
species variation in biogeochemical traits. In live vegetation, N, P, K, Mg, and $SO_4$ are all
positively correlated (Fig. S2), indicating that leaf tissues are built in predictable
stoichiometric ratios, but the variation in the concentration of those nutrients among
sampling sites suggests that the availability of these elements in soils and bedrock
likewise varied. Ca was weakly correlated with N, P, K, Mg, and $SO_4$ in leaves. Ca
concentration is strongly affected by foliage age (Turner et al., 1977), which represents
another source of variation in the sampled foliage not controlled for in this study.
We found substantial differences in the foliar nutrient concentrations between
evergreen and deciduous plant types (Fig. 3). N, Mg, and $SO_4$ were found at higher
concentrations in broadleaf material relative to evergreen needles, probably due to higher
rates of metabolic activity and photosynthesis in broadleaf material during the summer
(Linzon et al., 1979). Leaves from *Alnus,* a nitrogen-fixing plant, contained greater N
(2.55-3.56%) relative to non-fixing species (Taylor et al., 1989). Leaf N concentrations
ranged from 0.75 to 1.89% in the evergreen conifer species (i.e., excluding *Larix*
*occidentalis*) (n=63) —which is slightly lower but consistent with (1.15±0.24%) a
previously reported global average of 1.21±0.01% N for needleleaf evergreens (n=
5558)(Kattge et al., 2011). These data suggest that nutrient variation between plant
functional types (i.e. evergreen needleleaf versus deciduous) is much greater than inter-
or intra- specific variation within PFTs. This supports the use of broadly-defined PFTs in
biogeochemical models of ecosystem processes.
*3.2 Fire legacies on nutrient availability*
We compared average elemental concentrations (i.e. C, N, Ca, K, and Mg) in
organic soil horizons taken from sample sites burned in 1910 or 1918 CE with sites that
were burned prior to 1901 CE (i.e., the start of the Fire Atlas published by Morgan et al.
(2014))(Fig. 1,Table 1) to investigate the long-term effects of wildfires on soil nutrient
availability. Soil samples from the organic horizon that were burned in the early 20[th]





century contained significantly less C on average than samples from sites that burned
prior to 1901 CE (t = -2.26, df= 9.60, p= 0.0484). In contrast, N (t = -1.46, df= 10.54, p=
0.1731) and other soil nutrient concentrations (Fig. 5; Ca, K, Mg not shown) did not
significantly differ between these two populations.
245       The significant difference in soil C suggests that a single fire event in the early
20$^{th}$ century at several sites reduced the pool of soil C, and that the legacies of those fires
are still detectable today (Fig. 5). These results are consistent with ecosystem modeling
experiments, which find that both fire frequency and severity are the dominant drivers of
C dynamics in sub-alpine coniferous forests over centennial to millennial timescales
(Hudiburg et al., 2017). Furthermore, the greater variance in C concentrations observed in
soils burned in the early 20$^{th}$ century ($\sigma^2$=37.31) versus soils not burned ($\sigma^2$=12.44)(Fig.
5) may also reflect aspects of fire history, and specifically high spatial variability in fire
severity and post-fire recovery.
254       The results from this study are broadly consistent with ecosystem model
simulations suggesting that changing fire regimes under climate change scenarios have
the potential to alter C stored in forested ecosystems by changing the frequency and
intensity of wildfire events (Hudiburg et al., 2017;Kelly et al., 2016). We show that even
in a relatively small region of subalpine forests in the Rocky Mountains, an assumption
of a single value for C stocks would not hold, and are instead highly dependent on a
spatially heterogeneous fire history extending back for a least a century. The importance
of this variability in determining post-fire C dynamics implies that equilibrium scenarios
extrapolated from a single fire event in one location are a poor assumption when
simulating fire regimes in Earth System models at spatial scales larger than an individual
site (Hudiburg et al., 2017).
*3.3 Evaluating nutrient limitations*
268       The process whereby forest vegetation typically returns to pre-fire conditions after
several years or decades following wildfire events is referred to as biogeochemical
resilience (McLauchlan et al., 2014;Smithwick, 2011). This resilience is determined, in
part, by the availability of growth limiting nutrients, such as N and P (Güsewell,
2004;Schreeg et al., 2014). In non-N$_2$-fixing plant species, foliar N and P concentrations
reflect soil N and P availability (Reich and Oleksyn, 2004). Previous studies have shown
that a N/P mass ratio in foliar material of about 10-20 is optimal for plant growth (Aerts
and Chapin, 1999;Ingestad and Lund, 1979). N/P ratios >16 can indicate P-limited
biomass productions, whereas N/P ratios <14 are suggestive of N-limited plant growth
(Aerts and Chapin, 1999;Koerselman and Meuleman, 1996).
278       The average (± 95% confidence intervals) N/P values for sampled foliage (n=72)
was 9.8 ± 0.6. This suggests a N-limited growing environment. Although N/P ratios of
leaves are often used to evaluate nutrient status in plants, several factors could complicate





this interpretation: (1) N/P ratios of old leaves may be different than new foliage growth
(Schreeg et al., 2014). When plants are older, nutrients are reallocated to active
meristems (e.g., young leaves, shoot tips) (Güsewell, 2004). We did not account for
foliage or plant age during sampling. (2) All foliage was recovered in the summer,
therefore N/P ratios could be low because samples were taken during the period of active
growth (Méndez and Karlsson, 2005;Rivas-Ubach et al., 2012). (3) As a result of N
limitation, biomass allocation to roots can increase at the expense of foliage (Andrews et
al., 1999;De Groot et al., 2003). (4) P uptake can be enhanced in response to deficiencies
through several mechanisms, such as exudation of enzymes or acids (Dakora and Phillips,
2002) and association with mycorrhizal fungi (Colpaert et al., 1999). (5) Productivity
might also be limited by other elements (van Duren and Pegtel, 2000), solar irradiance,
and/or climatic factors (Spink et al., 1998).
To assess potential nutrient limitations resulting from wildfire, we compared
average N and P concentrations in foliar material from plants grown in soils that were
burned in 1910 and 1918 CE (n= 29) with sites not burned (n= 43) since the start of the
Fire Atlas (Morgan et al., 2014). We found no significant difference in average N (t = -
0.13, df= 47.76, p= 0.8948) or P (t= 0.99, df= 53.48, p= 0.3258) concentrations between
the two sample groups. Likewise, average C (t= -0.40, df= 65.96, p= 0.6901)
concentrations did not significantly differ between foliar samples taken from burned
versus unburned locations. This indicates that, although past wildfires reduced the pool of
soil C, they did not affect the concentration of growth limiting nutrients measured in
vegetation a century later.
**4. Conclusions**
Analyses of foliar and soil samples from a network of sites located in the northern
U.S. Rocky Mountains indicate the spatial distribution of key elements in subalpine
forested ecosystems. The concentration of C and nutrients (N, K, Mg, Ca) in soils is
highest in the upper litter and organic horizons and decreases at depth in the mineral soil,
consistent with previous studies. Comparing the two plant functional types, needle leaf
plants contain higher concentrations of C, while broadleaf material is enriched in N and
other trace elements (Mg, $SO_4$). Sites that were burned in a regionally extensive wildfire
in the early 20[th] century contained significantly lower C on average in the organic soil
horizon, compared with sites that burned prior to the 20[th] century. This highlights the
important role of wildfires as a dominant driver of soil C dynamics in sub-alpine forests,
with legacies that can last for more than a century (Hudiburg et al., 2017). Furthermore,
the high degree of variance in soil C concentrations among burned soils is consistent with
the inherent spatial heterogeneity in fire severity seen in contemporary fires. This spatial
heterogeneity adds to additional complexity Earth Systems Modeling efforts to represent
fire across space and time. The low average values of foliar N/P ratios (9.8 ± 0.6) suggest



that N may be in low supply to plants. Therefore N availability in soils may play an
important role in understanding the biogeochemical resilience of coniferous forests to
wildfires. Finally, these data contribute empirical data to efforts to model the
biogeochemical consequences of wildfires in the subalpine forests of western North
America.
*Data Availability*
Foliar data are available in the TRY Plant Trait Database.
*Supplemental information*
Supplementary information is available online.
*Author contribution*
DPP and KKM designed the study, analyzed the data, and prepared the paper with
contributions from BC, KW, and PEH. KW and PEH selected study sites. KW, PEH, and
KKM collected the samples. BC conducted laboratory analyses to acquire data and
assisted with data analysis.
*Competing interests*
The authors declare that they have no conflict of interest.
*Acknowledgments*

This work was supported by NSF award DEB-1655179 to KKM, and DEB-1655121 to
PEH. DPP was supported by NSF DEB-1145815. K. Lowe, J. Roozeboom and A.
Marcotte provided analytical assistance. We thank other Big Burns team members T.
Hudiburg, K. Bartowitz, and B. Shuman for field assistance and valuable discussions.





Table 1. Study sites

| Site name | Elevation m asl | Latitude | Longitude | Year of last major fire (CE)[1] |
|---|---|---|---|---|
| Cliff | 1810 | 47.1390 | -115.1919 | 1910 |
| Heart | 1682 | 47.2856 | -115.4529 | unknown |
| Hoodoo | 1817 | 46.9789 | -115.0012 | unknown |
| Hub | 1819 | 47.2756 | -115.3757 | unknown |
| Kid | 1909 | 46.7782 | -114.8146 | 1910 |
| Little Montana | 2011 | 46.8151 | -114.7652 | unknown |
| Lost Lake | 1840 | 47.1047 | -115.1274 | 1910 |
| Lower Bonanza | 1922 | 47.0916 | -115.1330 | 1910 |
| Lower Oregon | 1811 | 47.0554 | -115.0913 | unknown |
| Missoula | 1807 | 47.0671 | -115.1162 | 1910 |
| Moore | 1629 | 47.1818 | -115.2521 | unknown |
| Silver | 1623 | 47.3596 | -115.5659 | 1918 |
| Surveyor | 1831 | 46.8241 | -114.7585 | unknown |
| Upper Bonanza | 1922 | 47.0916 | -115.1330 | 1910 |
| Upper Oregon | 1811 | 47.0554 | -115.0913 | unknown |

[1]This coverage includes fire perimeters recorded from 1901-2008 CE (Morgan et al., 2014).










Fig. 1) (Bottom) Map of the study area with 1910 CE fire extent polygons (Morgan et al.,
2014). (Upper right) Regional map of the study region.





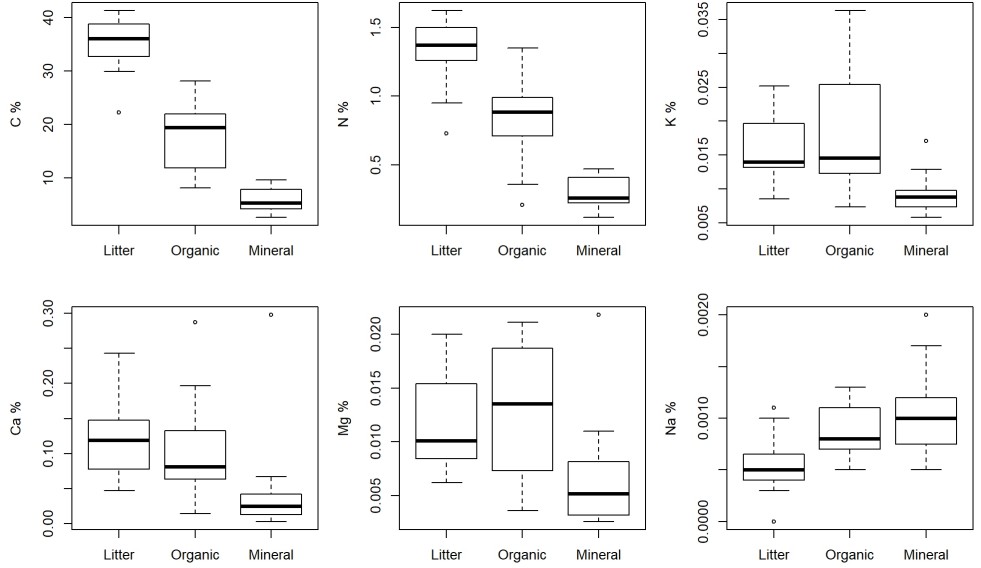

Fig. 2) Boxplots displaying element concentrations in soil samples.





Fig. 3) Boxplots displaying element concentrations in foliar samples. Subtypes are *Abies*
*lasiocarpa* (ABLA), *Alnus viridis* (ALVI), *Larix occidentalis* (LAOC), *Menziesia*





*ferruginea* (MEFE), *Pinus albicaulis* (PIAL), *Pinus contorta* (PICO), *Picea engelmannii*
(PIEN), *Taxus brevifolia* (TABR), *Thuja plicata* (THPL), and *Tsuga mertensiana*
(TSME).

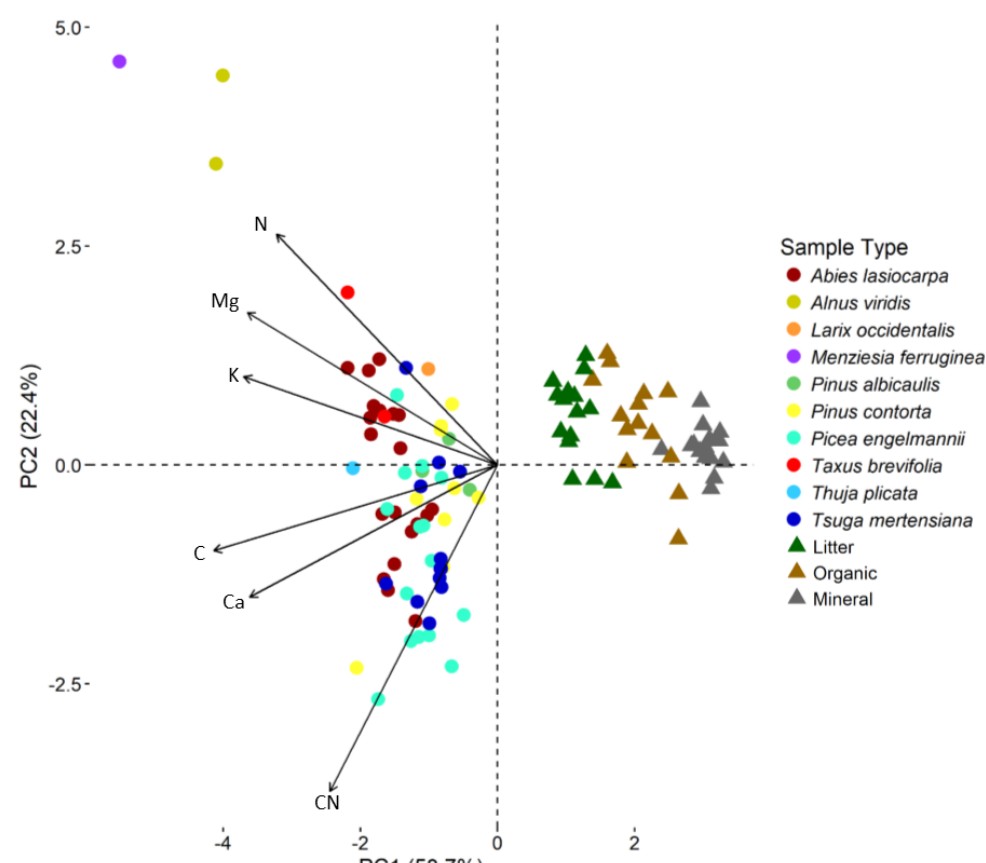

Fig. 4) Principal component analysis of soil and vegetation element concentrations.



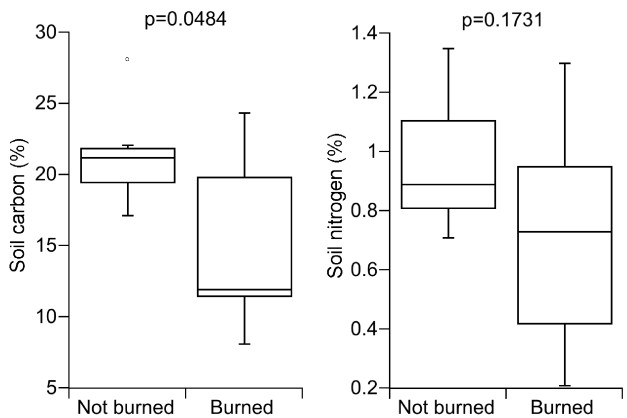

Fig. 5) Boxplots of C and N from the organic soil horizon burned (n= 7) in 1910 CE and
1918 CE versus sites not burned (n= 7) during the historical record (Morgan et al., 2014).
An unpaired t-test indicates that soil C concentrations are significantly (p=0.0484) lower
in burned sites.

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
