# Peer review of "Variation of key elements in soils and plant tissues in subalpine forests of the northern Rocky Mountains, U.S.A."

_Biogeosciences, 2018_

## Referee Comment (RC1) · Anonymous Referee #1 · 12 Dec 2018

There are difficulties to read. The title does not specify the temporality of the study and that it focuses on the long term. Authors do not speak at all of the long-term effect of fires on the ground. Just at the begginning of the abstract you analyze the effect of wildfires in these key soil elements. Please modify your title according to your study including the role of "wildfires" in these key elements. And also something to long-term. The introduction is tedious and unstructured in some parts as in the last paragraph. Without a structural logic or coherence in the "n" used in each case that varies from samples to subsamples and according to the elements analyzed. They provide information on soil properties that have not been analyzed. Missing information about the study area. Failures in the experimental design, depth of sampling, without

references to other papers with the same methodology. The latter also occurs with laboratory analyzes where the description finds lack of scientific support and is chaotic. The MS must be bounded according to what is to be analyzed (the topic of the study) and from there to do it. Different depths, vegetation, years from the last wildfire are mixed, probably different soil type due to the length of the transect and the differences in the vegetation, areas of different severity of fire are mixed without knowing the pre-fire conditions and where it is assumed that it was of high severity but it is not said why and due to the heterogeneity of the severity in a fire, by the pulses of the fire, the authors try to synthesize without success the effects on which said severity depends but they do not detail in their study how the severity was in this case. They use ratios like the N / P without having analyzed P. It does not follow the same order of description of the elements in all the sections of the paper.

Specific Comments Material and Methods Please provide information in "Study area" section about topography, soil type according to SSS or WRB, slope, aspect, mean anual temperature and precipitation, recent wildfires, etc. Lines 83-90 should be placed in "Study area". Authors appointed that there are 15 sampling areas but in line 103 appointed that there are 14. Line 113: Why id you selected this depth despite the low termal conductivity of soil?? Please, add references where this experimental designed was used to check the scientific validation of your study. If you are study areas where the last wildfire was 100 years ago, why did you sample to 25 cm? Line 114: How many cores did you selected form each site? and how did you select the studied sites? Lines 125 and 126: Then, as I can understand, you only took one sample from 0-10 cm depth. Why? This depth vary in each core? How you can know if the selected soils are from previously or after a wildfire? How many samples did you analyze from each depth? Line 143: So then you did not use n=44? Please, clarify here and in study Fieldwork section how many samples you took from each area, depth, etc. Line 146: As I can underestand, then, all the soil elements do not have the same "n" value. Please clarify. Results and discussion The discussion is scarce and poorly focused on MS topics. Conclusions The conclusions are appropiated to the MS.

---

## Referee Comment (RC2) · Anonymous Referee #2 · 4 Jan 2019

This study examined the variation of some element concentrations of foliage and soils collected from 15 sites in the northern Rocky Mountains, USA. The results showed that the foliage C and N concentrations differed between broad-leaved and conifer trees, and soil C concentrations differed between not-burned and burned sites. I think that the present results are important as the basis to understand biotic and abiotic factors affecting the biogeochemical processes. However, there are a number of issues to be addressed before a recommendation could be made for publication in Biogeosciences.

1. The rationale and goal of this study are unclear. It should be required to mention what a kind of questions remain unclear, why and how the question is important, and

how this study is designed to solve the question. For example, why was it necessary to examine the spatial variability of the elements of soil and plants? I suppose that the effects of wild fire on elemental composition could be the main topic for this study. It would be required to reorganize largely this manuscript to clarify these points.

2. The authors would need to explain about materials and methods more carefully. For example, it is unclear how many soil cores were collected from each study site, and how many plant individuals of the same species were examined at each study site. How were the samples dried (L131)? For what the authors made the comparison (L153)? In particular, it is not clear to me why the authors applied PCA for the element compositions of soil and plants. Was it important to demonstrate the difference in elemental compositions between soil and foliage? Please clarify.

3. I would like to recommend that Results and Discussion section (L159) would be separated into Results section and Discussion section in order to present clearly the present findings and the interpretation.

---

## Author Comment (AC1) · 20 Feb 2019

There are difficulties to read. The title does not specify the temporality of the study and that it focuses on the long term. Authors do not speak at all of the long-term effect of fires on the ground. Just at the begginning of the abstract you analyze the effect of wildfires in these key soil elements. Please modify your title according to your study including the role of "wildfires" in these key elements. And also something to long-term.

*We added "fire-affected" to the title. However, because we did not conduct a long-term study, we did not add the phrase "long-term" to the title. All samples were obtained in the period of a month. While we do investigate the past fire history (the early 20th century) of the sites, we did not conduct "long-term" research.*

The introduction is tedious and unstructured in some parts as in the last paragraph.

*We edited the introduction and last paragraph to simplify the text and explicitly state some of the key unknowns in the literature that we address with the study. In the third paragraph we added a new opening sentence to improve the transition. We revised the last paragraph to clearly state the objectives of the research.*

Without a structural logic or coherence in the "n" used in each case that varies from samples to subsamples and according to the elements analyzed.

*The use of "n" in the manuscript indicates the sample sizes used to calculate averages or confidence intervals. In the places where "n" may be ambiguous, we added figure calls to provide the source of our data in detail (see supplemental information). In line 120-121 we explain the discrepancies between "n" sample sizes between soil cores and vegetation samples. Our entire dataset is found in the main text and supplemental material.*

They provide information on soil properties that have not been analyzed.

*We are unsure what the reviewer means by this statement. We only describe the properties of soils in our results and discussion that have been analyzed and presented in Table 1 and S1. We welcome any clarification from the reviewer.*

Missing information about the study area.

*It is unclear what information is missing from the study area. We welcome suggestions from the reviewer as to what information should be added regarding the study area. We did add information on modern climate conditions, adding detail to the previous description.*

Failures in the experimental design, depth of sampling, without paper references to other papers with the same methodology.

*It is unclear what failures the reviewer is referring to with respect to experimental design. Due to variable depths of soils at the study sites, we sampled soil layers based on visual characteristics described in lines 128-131.*

The latter also occurs with laboratory analyzes where the description finds lack of scientific support and is chaotic. The MS must be bounded according to what is to be analyzed (the topic of the study) and from there to do it. Different depths, vegetation, years from the last wildfire are mixed, probably different soil type due to the length of the transect and the differences in the vegetation, areas of different severity of fire are mixed without knowing the prefire conditions and where it is assumed that it was of high severity but it is not said why and due to the heterogeneity of the severity in a fire, by the pulses of the fire, the authors try to synthesize without success the effects on which said severity depends but they do not detail in their study how the severity was in this case.

*We have added or highlighted text to better frame and describe the study. Soils are consistent across the study area and are characterized as shallow entisols (line 108). Soil horizon depths for sampling were determined visually in the field (lines 129-131). The years since the last known wildfire are listed in Table 1. Since the last fire in the study area occurred more than 100 years ago, detailed information about the degree of spatial heterogeneity of the fire is unknown. Therefore we relied on the published Fire Atlas from this region to determine the fire boundaries (see references). Additionally, we attempted to characterize pre-fire conditions by utilizing sampling sites where there have been no documented fires in the last ~110 years.*

They use ratios like the N / P without having analyzed P. It does not follow the same order of description of the elements in all the sections of the paper.

*We did analyze P. Specifically, we analyzed N and P in vegetation, which is stated in the methods and listed in Table S2. Since we structured the manuscript following the stated objectives in the introduction, we feel it does not make sense to restructure the paper around the presentation of elemental concentrations.*

Material and Methods Please provide information in "Study area" section about topography, soil type according to SSS or WRB, slope, aspect, mean anual temperature and precipitation, recent wildfires, etc.

*The soil types in the study area based on SSS (i.e. entisols)(line 108). Slope and aspect were added to Table 1. We added climate information (1912-2016 CE) from the nearby Western Regional Climate Center station at Haugan, Montana (station # 243984). Information about historical wildfires is included in Table 1.*

Lines 83-90 should be placed in "Study area".

*We feel that this section should remain in the introduction because it states the objectives of the manuscript.*

Authors appointed that there are 15 sampling areas but in line 103 appointed that there are 14.

*We thank the reviewer for catching this error. This was fixed in the text (line 103)*

Line 113: Why id you selected this depth despite the low termal conductivity of soil?? Please, add references where this experimental designed was used to check the scientific validation of your study.

*Soil cores were approximately 25 cm long, depending on soil depth. Due to low thermal conductivity, we sampled soil horizons at three depths described in lines 127-131. This included litter and shallow organic soil horizons (<10 cm depth). This method is commonly used across many areas of soil research as described in the Standard Soil Methods for Long-Term Ecological Research textbook (Robertson et al. 1999), which we added to the methods text.*

If you are study areas where the last wildfire was 100 years ago, why did you sample to 25 cm?

*We are slightly unsure of the reviewer's question. We applied commonly used soil sampling methods, sampling three depths in the soil core based on visual identification of the horizons: litter (unconsolidated plant material at the surface), organic (<10 cm), and mineral soils (10-25 cm depending on soil depth). The mineral horizon was analyzed to provide information on parent material contributing to soil formation.*

Line 114: How many cores did you selected form each site? and how did you select the studied sites?

*One soil core was selected from each sample site (line 120). Study sites were selected as part of an ongoing study that is also investigating lake-sediment records; all sites are adjacent to a lake in subalpine forest (selected by elevation and forest type). The study area has a history of fire activity, throughout the 20th century and prior (e.g., known from lake-sediment records). Individual sampling locations at each lake were selected randomly from within 100 m of the lake edge.*

Lines 125 and 126: Then, as I can understand, you only took one sample from 0-10 cm depth. Why? This depth vary in each core? How you can know if the selected soils are from previously or after a wildfire? How many samples did you analyze from each depth?

*The organic horizon was identified visually (line 130) and sampled at varying depths between 0-10 cm based on the thickness of the horizon at the core site. Soils were not sampled based on an assessment of their origin before or after a wildfire; rather, they were sample based on the visual transitions between horizons. We analyzed soil cores from all 15 sampling sites. At sites with soil cores, we analyzed one sample per soil*

*horizon (for a total of three, one each for each of three horizons) for multiple variables described in the methods section. It should be noted that only two horizons were analyzed at the Upper Oregon sample location because it did not contain a preserved organic soil horizon (see Table S1).*

Line 143: So then you did not use n=44? Please, clarify here and in study Fieldwork section how many samples you took from each area, depth, etc.

*It is unclear what sample size the reviewer is referring to since "n=44" is not found in line 143. We included all data used in our analysis in Table 1, Table S1, and Table S2. We took one sample from each of the three identified horizons in the soils cores described in lines 129-131 (except Upper Oregon, see table S1). All foliage samples taken from each sampling site can be found in Table S2.*

Line 146: As I can underestand, then, all the soil elements do not have the same "n" value. Please clarify.

*It is unclear what the reviewer is referring to in this comment. All soil samples were analyzed for the same variables. We state in line 146 that soils were not analyzed for P.*

Results and discussion The discussion is scarce and poorly focused on MS topics. Conclusions The conclusions are appropiated to the MS.

*We improved the discussion to focus more on the manuscript topics. The results and discussion section are now organized to correspond with the objectives of the study clearly described in the introduction.*

---

## Author Comment (AC2) · 20 Feb 2019

1. The rationale and goal of this study are unclear. It should be required to mention what a kind of questions remain unclear, why and how the question is important, and how this study is designed to solve the question. For example, why was it necessary to examine the spatial variability of the elements of soil and plants? I suppose that the effects of wild fire on elemental composition could be the main topic for this study. It would be required to reorganize largely this manuscript to clarify these points.

*We thank the reviewer for this helpful comment. We specified problems and shortcomings in the literature that are addressed by the study (lines 48, 67, 100). We clarified in the title and introduction that one of the main objectives of the study is to determine the legacy of fires on the availability of nutrients in soils and vegetation (lines 72, 100, 109).*

2. The authors would need to explain about materials and methods more carefully. For example, it is unclear how many soil cores were collected from each study site, and how many plant individuals of the same species were examined at each study site.

*All data used in the study are listed in Tables 1, S1, and S2. We took one soil core from each sampling location (line 144).*

How were the samples dried (L131)?

*Soils were dried in an oven at 60° C for 24 hours. This was clarified in line 136.*

For what the authors made the comparison (L153)? In particular, it is not clear to me why the authors applied PCA for the element compositions of soil and plants. Was it important to demonstrate the difference in elemental compositions between soil and foliage? Please clarify.

*We compared elements in soils and vegetation to provide quantitative information regarding patterns in concentrations between the sampling sites, and to evaluate the legacy of past wildfire impact on the distribution of elements. We used principal component analysis to differentiate elemental signatures in foliage and soils (lines 208-210). We clarified the text by presenting the principal component loads in a new table (Table 2). We feel PCA was important to help characterize patterns in elemental distribution among soil and foliage.*

3. I would like to recommend that Results and Discussion section (L159) would be separated into Results section and Discussion section in order to present clearly the present findings and the interpretation.

*We find that our results and discussions are better presented in the same section. This avoids redundancy as our datasets are available in the supplemental information. By combining the sections, we can discuss the implications of the data without having to re-state the main patterns in the text.*